# Fear of Cancer Recurrence among Aboriginal and Torres Strait Islander Women Diagnosed with Breast Cancer

**Tamara Butler [1,2,\*], Ben Smith [3,4], Kirsten Pilatti [5], Bena Brown [1,2], Kate Anderson [1,2], Bronwyn Morris [1,2] and Gail Garvey [1,2]**

1. School of Public Health, Faculty of Medicine, The University of Queensland, Herston, QLD 4029, Australia
2. Wellbeing and Preventable Chronic Diseases Division, Menzies School of Health Research, Charles Darwin University, Casuarina, NT 0810, Australia
3. Faculty of Medicine and Health, South West Sydney Clinical Campuses, University of New South Wales (UNSW Sydney), Liverpool, NSW 2170, Australia
4. Ingham Institute for Applied Medical Research, Liverpool, NSW 2170, Australia
5. Breast Cancer Network Australia, Melbourne, VIC 3124, Australia
\* Correspondence: t.butler@uq.edu.au

**Abstract:** Little is known about the fear of cancer recurrence (FCR) severity, coping strategies, or quality of life impacts for Indigenous populations. This mixed-methods study aimed to (1) quantify FCR levels among Indigenous Australian (i.e., Aboriginal and Torres Strait Islander) breast cancer survivors and (2) qualitatively explore experiences of FCR and the coping strategies used. Nineteen participants completed the Fear of Cancer Recurrence Inventory (FCRI); ten also completed a semi-structured interview. Interview transcripts were thematically analysed. Average FCR levels (Mean FCRI Total Score = 71.0, SD = 29.8) were higher than in previous studies of Australian breast cancer survivors, and 79% of participants reported sub-clinical or greater FCR (FCRI-Short Form $\geq$ 13/36). Qualitative themes revealed the pervasiveness of FCR, its impact on family, and exacerbation by experience/family history of comorbid health issues. Cultural identity, family, and a resilient mindset aided coping skills. Greater communication with healthcare providers about FCR and culturally safe and appropriate FCR care were desired. This study is the first to assess FCR among Aboriginal and Torres Strait Islander breast cancer survivors, extending the limited literature on FCR in Indigenous populations. Results suggest FCR is a significant issue in this population and will inform the development of culturally appropriate interventions to aid coping and improve quality of life.

**Keywords:** fear of cancer recurrence; cancer; oncology; Aboriginal and Torres Strait Islander; indigenous; cancer survivorship; supportive care; quality of life; coping mixed methods

## 1. Introduction

Fear of cancer recurrence (FCR) is a debilitating condition affecting cancer survivors defined by the experience of 'fear, worry, or concern relating to the possibility that cancer will come back or progress' [1]. While it is regarded as normal for cancer survivors to experience some degree of FCR, current estimates suggest that more than half (58.8%) of cancer survivors experience clinically significant FCR [2] that can persist for many years post-treatment [3]. Notably, the condition has been reported as one of the chief concerns of cancer survivors [3], and FCR is associated with adverse outcomes, including psychological distress; impaired social functioning, mental abilities, and coping with work; reduced quality of life and level of enjoyment; and increased healthcare usage and costs [4–6]. Recent advances have been made in FCR screening [7], assessment [8], and the development of efficacious treatments for FCR in some patient populations [9,10].

Little is known about FCR among Aboriginal and Torres Strait Islander cancer survivors. Compared with other Australians, Aboriginal and Torres Strait Islander people experience significantly poorer overall cancer outcomes with respect to risk factor prevalence,

cancer incidence, and disease outcomes [11–13], including psychosocial outcomes [14,15]. Breast cancer is the most commonly diagnosed cancer among Aboriginal and Torres Strait Islander women [13]. Although Aboriginal and Torres Strait Islander women are slightly less likely to be diagnosed with breast cancer than other Australians, they are more likely to die from the disease. Aboriginal and Torres Strait Islander women diagnosed with breast cancer have a lower five-year relative survival rate (81%) compared with other Australians (90%) [13]. Higher FCR is associated with female sex and younger age, so breast cancer survivors may be particularly vulnerable to FCR [16]. The scarcity of research into FCR among Aboriginal and Torres Strait Islander people makes it difficult to establish if and how FCR is experienced by Aboriginal and Torres Strait Islander breast cancer survivors and how best to support them in managing FCR.

A recent systematic review of FCR among Indigenous, ethnic, and racial minority populations highlighted considerable variability in the prevalence, lived experience, and factors associated with FCR across groups. However, only one study reported on FCR in any Indigenous population, namely Native Hawaiians [17]. These findings reinforce the need to better understand FCR specifically for Aboriginal and Torres Strait Islander breast cancer survivors, as there may be differences in FCR triggers, presentations, experiences, and potential treatments for this population.

Using a mixed-methods approach, this study is the first to explore FCR among Aboriginal and Torres Strait Islander women diagnosed with breast cancer. This study aimed to (1) quantify levels of FCR among Aboriginal and Torres Strait Islander women diagnosed with breast cancer and (2) gain a qualitative understanding of FCR and explore coping strategies to manage FCR used by Aboriginal and Torres Strait Islander women.

## 2. Materials and Methods

### 2.1. Research Team Reflexivity Statements

TB is a female Aboriginal early career researcher experienced in qualitative research with Aboriginal and Torres Strait Islander peoples with an interest in reducing health inequities in cancers that affect Aboriginal and Torres Strait Islander women and families.

BS is a non-Indigenous male research fellow with experience in qualitative and quantitative research focused on ensuring access to high-quality, evidenced-based, and culturally sensitive care for FCR for those living with and beyond cancer.

KP is a non-Indigenous woman and CEO of Breast Cancer Network Australia, which, with support from its First Peoples Advisory Group, provides information and resources tailored to meet the needs of Aboriginal and Torres Islander women with breast cancer.

BB is a non-Indigenous female clinician and researcher focused on person-centred care qualitative research and delivery of cancer care in Indigenous and non-Indigenous populations.

KA is a non-Indigenous Australian female senior research fellow experienced in conducting collaborative qualitative research with Indigenous researchers and communities.

BM is a non-Indigenous female researcher with experience in mixed methods research focussed on supportive care after cancer.

GG is an Aboriginal senior research fellow with experience in qualitative and quantitative research and an interest in achieving health equity for Indigenous peoples through a focus on cancer and well-being research.

### 2.2. Study Design & Participants

This study employed a mixed methods design. Eligible participants were women diagnosed with breast cancer; who identified as Aboriginal and/or Torres Strait Islander; were aged 18 years or over at the time of recruitment; and who had volunteered to be contacted by Breast Cancer Network Australia (BCNA) for purposes such as research participation.

### 2.3. Procedure

Reporting of qualitative methods and analysis is in line with COREQ [18]. Purposive sampling was used to identify individuals using a pre-existing BCNA database, including

299 women diagnosed with breast cancer who identified as Aboriginal and/or Torres Strait Islander. A BCNA representative invited eligible women to participate via email. Several reminder emails were sent, and some included video testimonials from Elders involved with BCNA sharing their breast cancer stories and encouraging women to participate. Social media posts by BCNA were also used for recruitment. The email invitation and social media posts included a brief overview of the project and links to the Participant Information and Consent Forms. Following eligibility checking and consent, participants proceeded to the Qualtrics (www.qualtrics.com, accessed on 12 October 2022) online survey, which remained open for approximately eight months.

The online survey included questions regarding demographics (e.g., age, marital status, education level, employment status, chronic conditions, etc.), cancer-related information (e.g., details of breast cancer diagnosis and status, treatment received, etc.), and the Fear of Cancer Recurrence Inventory (FCRI). The FCRI is a multi-dimensional measure containing 42 items grouped into seven subscales (triggers, severity, psychological distress, functioning impairments, insight, reassurance, and coping strategies; see Table 1) that together explain 64% of the variance in FCR [8]. Participants responded to FCRI questions using Likert scales from 0 ('never' or 'not at all') to 4 ('all the time' or 'a great deal'). One item ('I believe that I am cured and that the cancer will not come back') was reverse scored. The FCRI has a possible score range of 0–168, with higher scores indicating a greater FCR.

**Table 1.** Example items from the Fear of Cancer Recurrence Inventory and semi-structured interview.

| FCRI Domain | FCRI Item Example [8] | Qualitative Interview Guide Example Question |
|---|---|---|
| Severity<br><br>Psychological distress<br><br>Insight | I am worried or anxious about the possibility of cancer recurrence.<br>When I think about the possibility of cancer recurrence, I feel: worry, fear, or anxiety.<br>I feel that I worry excessively about the possibility of cancer recurrence. | • Have you ever had any worries about the possibility of your cancer getting worse or returning?<br>• Can you tell me a little bit about your worries?<br>• How often and for how long have you had these kinds of worries or thoughts? |
| Triggers | The following situations make me think about the possibility of cancer recurrence: television shows or newspaper articles about cancer or illness. | • What makes these worries or thoughts worse? |
| Reassurance<br><br><br>Coping | When I think about the possibility of cancer recurrence, I use the following strategies to reassure myself: I call my doctor or other health professionals.<br>When I think about the possibility of cancer recurrence, I use the following strategies to reassure myself: I try to distract myself. | • What makes you less worried about your cancer getting worse or returning?<br>• What strategies do you do to reassure yourself?<br>• Have you received any support relating to fear of cancer recurrence—for example, through support groups? |
| Functioning | My thoughts or fears about the possibility of cancer recurrence disrupt my work or everyday activities. | • Have you ever found that worries about cancer recurrence have impacted your day-to-day life?<br>• How does fear of cancer recurrence impact your family and other relationships? |

The final survey question invited participants to complete a telephone interview to further explore FCR. FCRI scores (total and subscale) were calculated prior to follow-up interviews to guide the discussion.

Interviews were conducted by experienced female researchers (TB, KA, and BB). The researchers contacted participants to reconfirm consent to participate in the interview and provide details on the purpose of the interview. An interview protocol (see Table 1) was used to guide the interviews and ensure rigour between interviewers. The interviews included open-ended questions exploring participant experiences of FCR around the seven FCRI subscales. Interviews were audio recorded using a digital voice recorder (Olympus Digital Voice Recorder WS-833) and transcribed verbatim by a professional transcription

service. On average, interviews took 37 min to complete. Participants were invited to review their interview transcript and edit or add comments, but none chose to do this.

Due to the small potential participant pool for the study, qualitative data collection ceased after ten interviews were conducted over a period of approximately five months. Data saturation had been reached at this point, with no new themes emerging in the final few interviews.

### 2.4. Analysis

2.4.1. Survey

Descriptive statistics for the demographic, cancer-related information, and each FCRI subscale were computed using SPSS version 27 (2020). Participants' residential postcodes were mapped to the Australian Bureau of Statistics's (ABS) Australian Statistical Geography Standard (ASGS), which categorises remoteness into very remote, remote, outer regional, inner regional, and major cities. These were further collapsed into remote, regional, and major cities [19].

2.4.2. Interviews

Transcripts were managed using NVIVO software version 12. Thematic analysis [20] was conducted based on Mertens' transformative approach [21]—one that privileges the voices of marginalised groups to reduce power imbalances and inequities. Privileging Indigenous voices and gathering culturally specific expressions, understandings, and knowledge of their circumstances is critical when conducting research with Aboriginal and Torres Strait Islander peoples. Three coders analysed the qualitative data (GG, TB, and BB). Coding was initially deductive, using the FCRI subscales as a guide, and then inductive to identify new themes from participant data. All three coders initially read three interviews to gain familiarity, iteratively meeting to reach a consensus on the application of coding. TB and GG's interpretation of data as Aboriginal researchers was privileged in consensus findings. Once consensus was reached between all three coders, TB and BB independently coded the remaining interviews. Following coding, the entire research team met to review themes, sub-themes, and exemplar quotes to gain group consensus.

### 2.5. Ethical Approval

Ethics approval for this research was obtained from the Human Research Ethics Committee (HREC) of the Northern Territory Department of Health and Menzies School of Health Research (2020-3848) and the University of Queensland HREC (2021/HE002400).

## 3. Results

### 3.1. Fear of Cancer Recurrence Inventory

Of the 23 participants who started the survey, 3 did not complete the eligibility questions, and 1 completed the demographic information but not the FCRI. All 4 were excluded from the dataset, leaving 19 participants. Table 2 shows the sample demographics and cancer-related information. Most women (94.7%) identified as Aboriginal, and the median age was 52 years (Mean (M) = 51.65, Standard Deviation (SD) = 9.51, range = 34–71). Many women were married (36.8%), working full-time (36.8%), and had completed vocational training (TAFE Diploma or certificate) (36.8%) as their highest level of education. Most women lived in the state of Queensland (52.6%) and in major cities (78.9%). The median number of chronic conditions was 2 ($M = 2.45$, $SD = 1.37$, range = 1–5).

Table 2. Survey participant characteristics and cancer-related information.

| Variable | Response Option | *N%* |
|---|---|---|
| Marital status | Never married | 4 (21.1) |
| | Widowed, divorced, or separated | 7 (5.3) |
| | Married/Defacto | 8 (36.8) |
| Highest level of education | No formal education | - |
| | Junior high school (Grade 10 or below) | 2 (10.5) |
| | Senior high school (Grade 11 or 12) | - |
| | TAFE Diploma/certificate | 7 (36.8) |
| | University degree | 8 (31.6) |
| | Other | 2 (10.5) |
| Employment | Full-time | 7 (36.8) |
| | Part-time | 6 (26.3) |
| | Unemployed | 2 (10.5) |
| | Retired | 3 (15.8) |
| | Other | 1 (5.3) |
| State/Territory | Queensland | 10 (52.6) |
| | New South Wales | 2 (10.5) |
| | Victoria | 3 (15.8) |
| | South Australia | 2 (10.5) |
| | Western Australia | 2 (10.5) |
| Geographic remoteness | Major City | 15 (78.9) |
| | Inner/Outer Regional | 3 (10.5) |
| | Remote/Very remote | 1 (5.3 |
| Language spoken at home | English | 12 (63.2) |
| | Aboriginal/Torres Strait Islander language | 6 (15.8) |
| | Other | 1 (5.3) |
| Children | Yes | 18 (94.7) |
| | No | 1 (5.3) |
| Number of chronic conditions | None | 8 (42.1) |
| | Two or less | 7 (21.1) |
| | Three or more | 4 (21.1) |
| Years since cancer diagnosis | 2 or fewer years | 6 (31.8) |
| | More than 2–5 years | 6 (31.8) |
| | 6–10 years | 5 (26.5) |
| | Greater than 10 years | 1 (5.3) |
| | Missing | 1 (5.3) |
| Cancer diagnosis | Ductal carcinoma in-situ (DCIS) | 3 (15.8) |
| | Early-stage breast cancer | 15 (78.9) |
| | Other | 1 (5.3) |
| Completed treatment | Yes | 13 (68.4) |
| | No | 5 (26.3) |
| | Missing | 1 (5.3) |
| Cancer treatments received [1] | Surgery | 14 (73.7) |
| | Radiotherapy | 14 (73.7) |
| | Chemotherapy | 11 (57.9) |
| | Endocrine therapy | 3 (15.8) |
| | Other treatment+ | 5 (26.3) |

[1] Participants may select multiple options; does not add to 100%.

Most participants reported an early-stage breast cancer diagnosis (78.9%), had no personal history of breast cancer prior to their current diagnosis (89.5%), and had been told they no longer had cancer (68.4%). The median time since diagnosis was 3.5 years (*M* = 4.4, *SD* = 3.6, range = 0.3–13.8).

Participant FCRI total and subscale scores are displayed in Table 3. The mean total FCRI score was 71.0 (*SD*= 29.8). Using a cut-off score of 22 or higher on the Severity subscale [22], almost half of the participants (*n* = 8, 42.1%) reported clinically significant FCR. About a third of participants (*n* = 7, 36.8%) reported sub-clinical FCR (severity subscale score 13–21).

Table 3. Fear of Cancer Recurrence Inventory scores.

| FCRI Subscale | Number of Items | Possible Range | Current Study (*n* = 19 Breast Cancer Survivors) | | | | | English FCRI Validation Study [23] (*n* = 140 Breast Cancer Survivors) | |
| --- | --- | --- | --- | --- | --- | --- | --- | --- | --- |
| | | | *M* | SD | Min | Max | α | *M* | SD |
| Triggers | 8 | 0–32 | 14.79 | 7.12 | 2 | 32 | 0.91 | 14.2 | 8.1 |
| Severity | 9 | 0–36 | 18.95 | 8.26 | 0 | 29 | 0.91 | 15.8 | 8.4 |
| Psychological distress | 4 | 0–16 | 7.05 | 4.31 | 0 | 12 | 0.84 | 5.1 | 4.3 |
| Coping Strategies | 9 | 0–36 | 17.42 | 7.30 | 0 | 24 | 0.88 | 16.4 | 9.5 |
| Functioning impairments | 6 | 0–24 | 6.84 | 7.22 | 0 | 23 | 0.96 | 3.6 | 5.6 |
| Insight | 3 | 0–12 | 1.74 | 2.81 | 0 | 9 | 0.76 | 1.1 | 2.1 |
| Reassurance | 3 | 0–12 | 4.21 | 3.33 | 0 | 10 | 0.78 | 2.5 | 2.8 |
| Total FCR | 42 | 0–168 | 71 | 29.84 | 14 | 120 | | 58.8 | 32.0 |

### *3.2. Qualitative Analysis*

All participants who completed the survey were invited to an interview. Ten Aboriginal women accepted the invitation, and all were interviewed. No reasons for declining the interview were recorded. Half reported clinically significant FCR, and three reported subclinical FCR in the preceding survey. The median age was 51.5 years (*M* = 53.8, *SD* = 10.6, range = 34 to 73), and most (*n* = 8, 80%) had been diagnosed with early-stage breast cancer and lived in major cities (*n*= 9, 90%). The median time since diagnosis was 2.8 years (*M* = 3.7, *SD* = 3.3, range 0.3–9.4). Over half of the interview participants (*n* = 6, 60%) had been told by their cancer care team that they no longer had cancer. Half had finished treatment, and the other half were receiving radiotherapy, chemotherapy, and/or endocrine treatments.

The qualitative analysis below is grouped by broad themes corresponding to FCRI sub-scales with further elaboration of sub-themes where applicable. Some new themes not covered in the FCRI were also generated.

### *3.3. FCRI Related Themes and Sub-Themes*
3.3.1. Psychological Distress, Severity, and Insight into FCR
Constant Source of Worry

Most participants experienced at least some FCR, which varied in severity, frequency, and duration. For most women, FCR was a constant worry and a deep source of stress after their breast cancer diagnosis and treatment. Fears included that the cancer would return to the breast, metastasise, or return in a different form.

> *[Cancer] is, it's just the most annoying thing ever. It's like this monkey on your back that just is never going to go away.* (P102)

Women often described feelings of uncertainty and were constantly questioning the long-term effectiveness of treatments and the accuracy of screening results.

> *It's like, hmm, okay, so we've got it [removed through surgery]. Yeah, yeah. So we've got it, but, like, okay, now what if it comes back?* (P105)

Two women were relatively unconcerned and were comforted by their treatment and monitoring of the cancer care team. Some participants described experiencing less FCR over time.

Impact of Worry on Family and Relationships

Women often discussed their worry about how cancer recurrence would negatively impact their families and relationships, above and beyond any concern for their own well-

being. Some worried about how their family would cope (both emotionally and practically) in the case of their death or illness.

*In the back of my mind, I just thought I couldn't go through this again. It was painful, the vulnerability, you know, everything and my kids seeing me go through this and everything.* (P103)

Women often worried they would not get to see significant milestones in their families' lives.

*So, you know, it's when I see my grandkids, you know, I think, "I hope I get to see your 21st." That sort of stuff.* (P106)

Women were also concerned family would eventually get cancer, and many had genetic testing to determine their own and family members' risk of cancer.

Some participants had an insight into their FCR.

*But when I'm on a good day [not worrying about FCR] it's like, yeah, I'm just doing this. I'm going through the motions and I don't feel scared. But it's probably when I'm on my bad days [when I think more about FCR].* (P106)

3.3.2. Predisposition to Cancer and Triggers

Family History of Cancer and Other Major Illnesses

A history of cancer in the family was a common predisposing factor that made women more likely to worry about their cancer returning. This included previous diagnoses of breast, as well as other cancers, such as bowel and lung. One woman who was adopted as a child worried about the potential unknown history of cancer in her family. A known family history of cancer sometimes prompted fatalistic thinking about their cancer prognosis.

*I reached out to family, and found that a cousin has had breast cancer, another aunty and my nanna had it, and it's just always in the back of my mind that, you know, how long do I have to live?* (P101)

Furthermore, women often reported that having family members with a major chronic illness, such as heart conditions, triggered fatalistic grief and concern that they would die of cancer.

*Because I'm the youngest of the five children. Like, my brother died of a heart attack last year, he was 59. And then the other second sister, the next one down, she had aneurisms a few years ago. So it's like my turn now, hey.* (P109)

One woman said her Aboriginal identity heightenedher perception of the risk of cancer recurrence.

*... being Aboriginal and having diabetes are the two standout ones [that I worry about] 'cause there's a very high rate of breast cancer in that particular category. [ ... ] just knowing that, um, for some reason Indigenous people are more commonly getting breast cancer.* (P105)

Physical and Mental Health Concerns

Other personal health concerns such as diabetes, skin cancer, and mental health triggered FCR and caused women to think they were at higher risk of recurrence than other women, or at the very least set off niggling worries.

*I've got diabetes and, um, a few other mental health problems and a few other physical health problems. And you sort of think about all those things put together, and you go, oh yeah, maybe that's why I'm tired. But it still enters your head that it just might be the cancer returning in a different form.* (P105)

Women were hypervigilant about changes in their physical health. Any general feelings of illness, pain, soreness, fatigue, breathlessness, or weight changes raised heightened concerns among women that their cancer was returning.

*And I've even only got to get the slightest headache and I'm like, "Oh, it could be a brain tumour. It might've come back in my brain."* (P105)

One woman reported a sudden increase in skin cancers following chemotherapy treatment, which she attributed to the possibility that her cancer had metastasised.

Furthermore, some women reported worries that stressful life events would cause the cancer to return. For example, one woman reported concerns that the stress caused by being in a violent relationship may have contributed to her cancer progressing, along with the stress caused by constant displacement and trauma.

Medical Appointments and Treatment

The anticipation of upcoming appointments, the appointments themselves, regular mammogram reminders, and side effects of treatment were common triggers of FCR. The time spent waiting for results between appointments was also filled with worry.

*And I think my worst time for thinking about all of that was just before my first chemo session . . . , you had to do your bone scan and your CT scan, and so just to check to see whether it's metastasised . . . I think was the worst week and a half ever, like I just worked myself up into such a state about it . . .* (P102)

*Oh, you hold your breath . . . Because I was 40, so over 10 years I've been having mammograms, because of my family history. And they show me them now, and I hold my breath, and I look at them, and I think, oh shivers, is that dark lump there, is that a brown spot there? You really stress out . . . my greatest fear is because I don't have a mum here anymore, it's for my little girl. What happens to her?* (P107)

Treatment for cancer and side effects caused FCR among some women due to the "brutal" (P014) nature of the experience and difficulties coping with treatment. For one woman, treatment paraphernalia triggered worry about FCR; she felt a "twinge of anxiety" seeing a topical anaesthetic patch due to its association with receiving treatment for cancer.

3.3.3. Functioning Impairment

Women discussed how FCR impacted their ability to function in various contexts, such as family and relationships, work, hobbies and leisure activities, and day-to-day life. One woman summed up the huge impact of FCR on her general functioning:

*It made a big impact on my whole life. I had to stop work for a little while, so that was really sad. I was too scared to look forward to things, you know, every time I tried to plan something, it just went crashing down. So I stopped looking forward to things and that's why I needed to see a psychiatrist, because it impacted my life. I found that cancer was taking over me, and I think my children noticed that and that was really sad.* (P103)

Family, Relationships, and Work

Most participants frequently thought about the negative impact their cancer diagnosis, treatment, and cancer recurrence would have on their family and loved ones. For some, it changed how much energy they had to interact with grandchildren; some worried about not being about to meet future grandkids or be present for significant milestones.

*Well, I've got grandchildren too. That sort of worries me, because I want to be around long enough to see them, their little milestones and that too. They're still having babies my children, and that worries—the whole thing, that not being around for them. Yeah, yeah. Being able to be a part of those significant life moments and that too.* (P109)

One woman worried about who would care for her children if she passed away. Another reported that, despite trying her hardest not to let FCR affect her, she had written cards with messages to her children in case she passed away.

One participant described feeling cautious about starting new relationships in case the cancer returned.

*I'm a little bit hesitant at starting any new relationships, like, with anyone . . . Like, even friendships. Ah, like, I want to, but part of me just says, "Oh well, I might be, you know, dead in six months from now, so what's the point? (P105)*

Some participants discussed that uncertainty about recurrence made them unable to continue work. Changes in work led to changes in finances, which caused stress.

*The not knowing whether it is still active in my body and that, until the treatments finished, I won't know. And that's 20–25 weeks of treatment, half a year. It's a long time. Like, I've had to resign from work because I was a nurse too, with the Aboriginal Medical Service."* (P109)

### 3.3.4. Reassurance and Coping
Support

Women received emotional, informational, and practical support from multiple sources to cope with FCR—family, friends, social media, psychologists, and support groups. Women often acknowledged that without support, coping with FCR would be quite difficult. Family was a particularly strong source of support.

*They keep my mind active, you know, I love my kids and grandkids and, yeah. So I was there for them and now they're, sort of, here for me.* (P101)

Talking about FCR and their cancer diagnosis was generally seen as a positive coping strategy, feeling that it helped validate their experience. However, one woman reported balancing seeking support from her husband with protecting him.

*When I was having a bad day, I would hide it from [my husband]. I wouldn't show it to him because he's so busy at work, and all that sort of stuff. And I just thought there's nothing he can do anyway . . . When it comes to support, if he hasn't supported me anywhere along the way it's because I've hidden something from him. Not because he can't or doesn't want to. I could not have wished for a better person.* (P106)

Some women gained strength by providing support to other Aboriginal and Torres Strait Islander breast cancer survivors by sharing their stories, giving talks, and sitting on the board of a breast cancer recovery group through a hospital.

Women proactively searched and found outreach services for counselling and support for FCR. This included state-based Cancer Councils, which offer various services to support people affected by cancer, including counselling, practical and emotional support, and support groups.

*So, I had a few fears, and I rang Cancer Council for some counselling. Ah, so there's a really good psychologist there. Um, yeah, she was very confident and, you know, could take a joke.* (P105)

Some sources of support were viewed with mixed feelings. Social media was sometimes seen as a "blessing and a danger for feeding your fears". (P102) Breast cancer support forums sometimes caused feelings of uncertainty when reading about others' cancer metastasising. Women avoided people who were not supportive or who encouraged toxic positivity.

*People tell me to think positive . . . I know they're trying to be really nice about it, but I don't feel like being positive . . . I'm sick of being positive, because every time I'm positive, I feel like I'm being let down, but I don't tell my friends that.* (P103)

Self-Examination, Screening, and Medical Appointments

Women took comfort and reassurance from regular self-examination, screening, and medical check-ups confirming that their cancer had not returned, although, for some, this was tinged with lingering fears that cancer would return.

*I check every lump, bumps, everything—every single day. Every time I have my scan, I'm thinking, am I going to get it again. Even like they found another lump in my mastectomy,*

*so my right side, I thought, how can I, they've taken it away. So it came clear, but you're still in the back of your mind, "Am I going to get it again?"* (P103)

Trust in Cancer Care Team, Medical System and Treatment

Some women's FCR was reduced by their steadfast faith in their medical team and the health care system, leading them to trust that the rigour and comprehensiveness of their treatment reduced the likelihood of recurrence.

*Rarely, rarely think about it ever coming back, because I think I've had enough radiation therapy and chemo and whatever to kill any cells.* (P110)

*. . . being able to be part of the solution with your doctors, you do what they say you have to do, and I've been very lucky where I've had really good doctors and it's worked out so far.* (P108)

However, for one woman, negative experiences with the medical system led to a distrust of doctors' decisions not to investigate symptoms.

*Ah, the doctor doesn't take me seriously when I say, "Look, I've got a headache." If they don't send me for a scan I just go see another doctor and get a scan . . . I just don't take the doctor's word for it anymore. [ . . . ] There's no law against having, you know, 10 doctors if you want to get the right answer and be—put your mind at rest, you know.* (P105)

Outlook and Attitude

Women displayed various mindsets regarding their cancer diagnosis and consequent FCR. Some maintained a positive outlook, while others had a very pragmatic approach of "getting on with it" by figuring out the next steps as needed and not dwelling on stresses.

*. . . that was a whole attitude I took through the whole process . . . it was just, right, let's get on with it . . . From day one, we both just said, right, we'll do whatever treatment that's needed. It's all about living, not about dying.* (P106)

*There is a possibility [cancer comes back] but I'm not going to allow it. That's my mind . . . .I'm just focused.* (P110)

Some women's positive mindset came from comparing their situation to others and emphasising what they could feel grateful for. For example, having supportive family and friends or being lucky that children were independent adults at the time of diagnosis.

Finally, a sense of humour sometimes helped women to cope with FCR, especially in conversations with the cancer care team. One woman reported joking with her oncologist about her cancer returning.

*I said, "Oh, well, I've only got two boobs, that's it." And he goes, ah, "It doesn't work that way," and I said, "I know that." He said, "It's really good that you're in good humour," and I said, "Oh well, maybe that's my way of, that's my mental health."* (P108)

Some women employed avoidance to cope with FCR. Some suppressed thoughts about it, and others used distraction (e.g., focusing on work, family, or study) to give themselves a break from worrying. Support from friends also helped with distraction.

*I think what takes the worry away is ignoring the fact that you've ever had it. You can forget it for the day.* (P105)

3.3.5. Need for Information and Resources

Some participants discussed that reputable information alleviated their FCR. Participants reported sourcing information independently of their cancer care team through internet searches, searching specific breast cancer and screening websites, trawling through social media group chats, and looking up other resources. People described a desire to learn more about cancer recurrence and their cancer diagnosis and that having more knowledge increased their awareness of what to look out for or prepared them for the potential side effects, while others reported this made their FCR worse.

*Yeah, researching about it and looking at reports and other—what other people have been through and—you know, one of the first things they tell you is, don't google anything. But it's the first thing that you want to do because you want to know more.* (P105)

Participants acknowledged the potential additional worries caused by seeking out more information on their risk of recurrence but that it was ultimately helpful to have this information.

*If you read the wrong thing it certainly does freak you out more, but, um, I'd say it'd be more helpful than not . . .* (P105)

Many women reported a general lack of information about the risk of cancer recurrence, and that information would have been helpful to support them in coping with their fears. One woman felt it would have been helpful to prepare her for her personal experience of cancer recurrence.

*I think [information on recurrence] would have helped me a lot, because I felt like I was on my own and I was feeling this on my own.* (P103)

One participant commented on the lack of culturally appropriate information resources—not only related to FCR but across the entire cancer journey.

*I mean, they give us this folder and it's all white orientated information . . . There's no culture to any of your journey.* (P109)

### 3.3.6. Communication and Care from Cancer Health Professionals

Participants discussed the importance of communication that is culturally responsive and the need for holistic, culturally appropriate cancer care. In follow-up appointments, women reported that the cancer care team was focused on the 'medical' side of cancer rather than the psychological or well-being aspects related to FCR, providing information on what would happen physically but nothing regarding cancer possibly recurring nor the impact on mental health.

*[Cancer care team usually asks about] how you've been going and the tablets I'm on. Usually I'll go in with a list of questions, but as for my anxiety and stress . . . I've never, sort of, brought that up with them, maybe I should but, yeah. . . . never had they asked about my actual wellbeing . . . Like, how am I coping and everything like that. It was only my GP, when I finally broke down and she said, "I've been waiting for this to come". And she got me onto the psychologist, she said, "You need to talk to someone".* (P101)

Some participants said their cancer care team had dismissed their expressed FCR.

*But whenever I would talk about it with people, with health professionals they go, oh, you've got no greater risk than anyone else. Don't worry about it, sort of, thing, and really just, sort of, brush it aside. . . . It made me feel, like, I was being a bit of a drama queen, or that I wasn't important sometimes too . . . she just completely discounted any fear that I had.* (P104)

### 3.3.7. Culture and Cultural Safety

Some participants reported drawing on culture as an Aboriginal woman to cope with FCR through practising art or visiting the Country.

*I'm looking actually to try and do some painting, so my Aunty said, "It's very therapeutic." So I've, sort of, done a few sketches, but I'm yet to go and get my paints and things like that. So I want to draw on my culture, I'm still learning about my culture . . .* (P101)

One participant raised several issues regarding the need to increase the Aboriginal and Torres Strait Islander clinical workforce to support women with FCR in a culturally safe way. While she appreciated being able to have a yarn (a conversation in a culturally safe environment) with Aboriginal and Torres Strait Islander Liaison Officers, she identified periods of time for education and health promotion opportunities and the need for more Aboriginal and Torres Strait Islander clinical staff.

> *. . . you have to wait on this Liaison Officer to come in who's so stretched in their time, to sit down and have a talk to you, and they can't talk to you really about your condition or what's going on because they're not a nurse . . .* (P102)

When asked what having access to Aboriginal and Torres Strait Islander clinical staff would have meant to her, she said:

> *. . . to be able to stop and have a bit of a yarn to someone . . . it really makes a difference. I'm trying to articulate it. I think there's a different kind of a connection that you get . . .* (P102)

Another participant commented on the limited availability of Aboriginal and Torres Strait Islander Liaison staff and the need for more Indigenous staff, as this would provide women with more support. This participant also spoke about the importance of having men's and women's Liaison officers to ensure Men's and Women's Business was respected.

## 4. Discussion

This mixed methods study is the first to quantitatively assess FCR in Aboriginal and Torres Strait Islander people and qualitatively explore its impact and related coping strategies. All FCRI subscale scores were higher than those reported by breast cancer survivors in the study validating the English version of the FCRI [23], but subscale scores assessing FCR-related distress and functional impairment were considerably higher, highlighting the negative impact of FCR on Aboriginal and Torres Strait Islander breast cancer survivors in particular. Correspondingly, FCRI data indicated that just under half (42%) of the participants reported clinically significant FCR, and a further third (37%) showed sub-clinical FCR. Clinical and sub-clinical FCR appear more prevalent in Aboriginal and Torres Strait Islander breast cancer survivors than in breast cancer survivors generally, 22% of whom report clinical FCR and 42% sub-clinical FCR. [2] Total FCRI scores ($M = 71.0$, $SD = 29.8$) were also slightly higher than other studies of Australian breast cancer survivors (e.g., $M = 68.0$, $SD = 25.6$) [24], similarly suggesting that Aboriginal and Torres Strait Islander breast cancer survivors are more severely affected by FCR. These findings are consistent with a systematic review of FCR in Indigenous and minority peoples, which documented differing FCR levels in majority populations [17]. Larger studies of Aboriginal and Torres Strait Islander cancer survivors are required to better estimate FCR prevalence and severity.

The qualitative component of the study found both similarities and differences in how Aboriginal and Torres Strait Islander women experience FCR relative to non-Indigenous cancer survivors. The experience of FCR was ubiquitous but varied in severity, similar to a meta-synthesis of qualitative FCR research, which found that FCR was a constant presence for many patients [25]. Ongoing uncertainty around potential recurrence was also noted as a challenge in the only previous study to touch on FCR in Indigenous (i.e., Native Hawaiian) people with cancer [26].

Women reported that heightened FCR was commonly triggered by imminent follow-up appointments and/or test results, similar to non-Indigenous cancer survivors [25,27,28]. However, in Aboriginal and Torres Strait Islander people, FCR linked to medical appointments may be exacerbated by experiences of institutional racism leading to mistrust of the medical system [29], as voiced by one woman who felt potential symptoms of recurrence were not taken seriously. Physical symptoms were also a commonly reported FCR trigger, as per non-Indigenous populations [3,27]. Aboriginal and Torres Strait Islander people are more likely to experience chronic illness due to inequalities in the social determinants of health, such as lack of access to healthcare, institutional racism, and socio-economic disadvantage. The higher prevalence of co-morbid chronic illnesses in Aboriginal and Torres Strait Islander cancer survivors is likely to make them more likely to experience physical symptoms and potentially be hypervigilant to bodily symptoms. This may further explain the higher FCR levels reported by Indigenous women.

FCR was experienced as distressing by Aboriginal and Torres Strait Islander breast cancer survivors, with the possibility of death or having to undergo further treatment particularly concerning, as in non-Indigenous populations [25,30]. While FCR is not

strongly associated with an objective risk of recurrence and prognosis in non-Indigenous populations [3], the poorer survival outcomes for Aboriginal and Torres Strait Islander women with breast cancer, and greater likelihood of experiencing life-limiting illness among family/community, may exacerbate FCR. Distress associated with FCR seemed linked to the impact a recurrence would have on their family. This has also been documented as a concern for non-Indigenous breast cancer survivors, particularly young mothers [25,31]. However, the central role of family and community in Aboriginal and Torres Strait Islander culture may mean that FCR is more strongly linked to concerns about the impact on their family.

Seeking social connection and support were reported as important FCR coping strategies. Sharing experiences of cancer and FCR with other cancer survivors, friends, and, in particular, family helped women validate concerns and obtain emotional support. Previous studies have noted the importance of social support from family to Indigenous cancer survivors [26], but this study is the first to find that social support aided coping with FCR specifically. Conversely, the importance of family and the desire to protect loved ones sometimes prevented breast cancer survivors from sharing FCR. Not sharing concerns with family and friends to protect them has been documented as a form of avoidant coping behaviour [25]. Caregivers do experience high levels of FCR [32], but not sharing concerns can actually exacerbate rather than protect against FCR [33]. Helping Indigenous breast cancer survivors to share their concerns with caregivers may help reduce the burden of FCR.

Women often proactively sought out information to aid in coping with FCR but noted some information (e.g., stories about recurrence) could increase anxiety, and it was sometimes hard to find information that was reputable and culturally appropriate. Taking an active approach to managing FCR has been noted as a common response in qualitative research [25], but the benefits of information-seeking may be hindered by limited information and support specifically for Indigenous people, which was also noted in Braun, Mokuau [26]. Women also indicated that follow-up appointments were often very medically focused, which meant that psychosocial issues like FCR affecting social and emotional well-being, which are important to the more holistic Aboriginal view of health [34], were often ignored. This is problematic, as patients often do not raise FCR for fear of seeming ungrateful or questioning the efficacy of treatment [35]. Aboriginal and Torres Strait Islander cancer survivors may be even less likely to raise FCR due to experiences of institutional racism within the healthcare setting, as noted above. Recruiting and training more Aboriginal and Torres Strait Islander healthcare staff, who better understand Indigenous patients' culture and perspective on health, may enable emotional concerns such as FCR to be raised and addressed in a culturally acceptable manner. Cultural safety training for non-Indigenous healthcare staff may also assist in this regard.

*Strengths and Limitations*

This paper responds to the call for further FCR research among Indigenous populations [17]. One previous qualitative study with Native Hawaiians briefly noted that FCR was common, but this is the first in-depth study focused on FCR in Indigenous cancer survivors, namely Australian Aboriginal and Torres Strait Islander women affected by breast cancer. Study strengths include that the research and analysis were led by Aboriginal and Torres Strait Islander women, which likely provided a sense of cultural safety and connection for participants and resulted in richer data and culturally informed analytical insights. The mixed methods approach also aided the evaluation of how the prevalence and severity of FCR in Aboriginal and Torres Strait Islander women with breast cancer compared to non-Indigenous populations as well as a greater understanding of factors underlying experiences of FCR and coping strategies among this population. The small sample limits the conclusions that can be drawn from the quantitative data but provides a foundation that can be built on by future research. Most qualitative study participants reported clinical or sub-clinical FCR, which may have meant more negative accounts of FCR experiences and coping strategies, although positive ways of coping with FCR were

also reported. Participants were also mostly from urban areas, so the potential challenges associated with living in regional or remote areas with fewer healthcare services did not feature prominently.

## 5. Conclusions

There is a dearth of research on Indigenous populations' experiences of FCR, including Aboriginal and Torres Strait Islander peoples in Australia. This is a significant concern given the greater burden of cancer among Indigenous populations. This study demonstrates that Aboriginal and Torres Strait Islander breast cancer survivors may also bear a greater burden of FCR. The findings provide a greater context and understanding of the experience of FCR among Aboriginal and Torres Strait Islander women with breast cancer, which is urgently needed to develop culturally appropriate and effective resources and psychosocial cancer care for this population.

**Author Contributions:** Conceptualization, K.P., B.M. and G.G.; methodology, G.G., B.M. and T.B.; formal analysis, G.G., T.B. and B.B.; investigation (interviews), T.B., K.A. and B.B.; resources, K.P.; data curation, T.B. and B.M.; writing—original draft preparation, T.B., B.S. and K.A.; writing—review and editing, T.B., B.S., K.P., B.B., K.A., B.M. and G.G.; supervision, G.G. (lead) and T.B. (secondary); project administration, B.M., G.G. and T.B.; funding acquisition, B.S. All authors have read and agreed to the published version of the manuscript.

**Funding:** This study was supported by the National Health and Medical Research Council (NHMRC) funded Centre of Research Excellence (CRE) in Targeted Approaches To Improve Cancer Services for Aboriginal and Torres Strait Islander Australians (TACTICS; #1153027). The views expressed in this publication are those of the authors and do not necessarily reflect the views of the funders. TB was supported by an Australian Research Council Discovery Australian Aboriginal and Torres Strait Islander Award (#IN190100050) and a National Health and Medical Research Council Investigator Grant (#2008097). BS was funded by the NSW Government through a Cancer Institute NSW Career Development Fellowship (#2021/DCF1138). BB and BM were partly funded by TACTICS (#1153027). GG was funded by a NHMRC Investigator Grant (#1176651).

**Institutional Review Board Statement:** Ethics approval for this research was obtained from the Human Research Ethics Committee (HREC) of the Northern Territory Department of Health and Menzies School of Health Research (2020-3848) and the University of Queensland HREC (2021/HE002400).

**Informed Consent Statement:** Informed consent was obtained from all subjects involved in the study. Written informed consent has been obtained from the patients to publish this paper.

**Data Availability Statement:** The data presented in this study are available on request from the corresponding author.

**Acknowledgments:** The authors wish to thank the participants involved in this research study. Ownership of Aboriginal and Torres Strait Islander knowledge and cultural heritage is retained by the informant.

**Conflicts of Interest:** The funders had no role in study design, data collection and analysis, the decision to publish, or the preparation of the manuscript.

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
