# Peer review of "Fear of Cancer Recurrence among Aboriginal and Torres Strait Islander Women Diagnosed with Breast Cancer"

_curroncol, doi:10.3390/curroncol30030222_

Round 1

Reviewer 1 Report

Quantitative part can be improved.

The main question addressed by the research  is the burden of cancer among Indigenous populations in Aboriginal and Torres Strait Islander peoples in Australia. Actually, we know very little about fear of cancer recurrence severity, coping strategies, and quality of life impacts in Indigenous populations.  Conclusion can be improved by describing the core outcome of the study. The references  are appropriate.

Reviewer 2 Report

In the manuscript titled 'Fear of cancer recurrence among Aboriginal and Torres Strait Islander women diagnosed with breast cancer', the authors conducted a mixed method approach to measure fear of cancer recurrence and associated coping mechanisms in specific population cohorts. Their findings suggest that there is a need to develop programs for culturally related interventions for improving quality of life including help in coping with fear. The study is sound and presents quality data and therefore can be recommended for publication.

Reviewer 3 Report

Dear authors,

I read with interest the article which analyzes the context and understanding of the  experience of FCR among Aboriginal and Torres Strait Islander women with breast cancer, which is needed to develop culturally appropriate and effective resources and psychosocial cancer care.

I agree with the authors that these problems must be addressed selectively to obtain important therapeutic results, while also protecting patient psycho-physical integrity.

Even if the number is small, it would be useful to discriminate the rate of conservative or demolitive surgery (with a possible percentage of prosthetic reconstruction). I ask the authors to implement this part if possible, thanks.

I found it well written, the English is good and the text is clear.

Even if the numbers are good, the techniques are not really innovative, but it is a interesting study especially considering the importance of this topic in the current practice.

I recommend this article for publication.

The manuscript clear, relevant for the field and presented in a well-structured manner.

The cited references mostly recent publications (within the last 5 years) are relevant.

The manuscript scientifically sound and is the experimental design appropriate to test the hypothesis.

The manuscript’s results reproducible based on the details given in the methods section (to be implemented).

The tables/ schemes are appropriate and they properly show the data (easy to interpret and understand).

The statistical analysis or data acquired are appropriate.

The conclusions are consistent with the evidence and arguments presented the ethics statements and data availability statements to ensure they are adequate.

In conclusion the question original and well-defined and the results provide an advancement of the current knowledge. In this regard the work fit the journal scope.

The results interpreted appropriately and are significant. All conclusions are justified and supported by the results and the hypotheses carefully identified. The article written in an appropriate way and the data and analyses presented appropriately.

The conclusions interesting for the readership of the journal and will the paper attract a wide readership.
